# Life Satisfaction and Perceived Stress versus Health Promoting Behavior among Medical Students during the COVID-19 Pandemic

**DOI:** 10.3390/ijerph19116706

**Published:** 2022-05-31

**Authors:** Karina Badura-Brzoza, Paweł Dębski, Patryk Główczyński, Małgorzata Dębska-Janus, Piotr Gorczyca

**Affiliations:** 1Department of Psychiatry, Faculty of Medical Sciences in Zabrze, Medical University of Silesia in Katowice, 42-612 Tarnowskie Gory, Poland; pdebski@sum.edu.pl (P.D.); pgorczyca@sum.edu.pl (P.G.); 2Institute of Sport Sciences, The Jerzy Kukuczka Academy of Physical Education in Katowice, 40-065 Katowice, Poland; m.debska@awf.katowice.pl

**Keywords:** COVID-19 pandemic, medical students, health-promoting behavior, life satisfaction, stress

## Abstract

Aim: The aim of this study was to evaluate health-promoting behaviors as a factor supporting life satisfaction and a protective factor against stress in a group of Polish medical students during the third wave of the SARS-CoV-2 pandemic. Material: The study was conducted in October–December 2021. There were 306 people, including 203 students in the 1st year of medicine, aged 20.42 ± 1.92 years, and 103 students in the 5th year of medicine, aged 24.88 ± 1.7 years. Methods: The following were used: the Satisfaction with Life Scale (SWLS), the Health Behavior Inventory (IZZ), and the Perceived Stress Scale (PSS-10). The survey was conducted online. Results: In the study group, the average result of 23.66 ± 5.97 was obtained in the life satisfaction questionnaire. In the overall assessment of health behaviors (IZZ), an average score of 80.06 ± 13.81 points was obtained. In the PSS-10 questionnaire, the respondents obtained an average of 21.10 ± 5.89 points. There were significant, negative correlations between the results obtained in the health behavior inventory with the results obtained in the stress scale and positive correlations with the results obtained in the life satisfaction scale. Moreover, in the assessment of the influence of prohealth behavior parameters on perceived stress and life satisfaction, a positive effect of PMA on life satisfaction was demonstrated, as well as a protective effect of Positive Mental Attitude (PMA) and Prohealth Activities (PhA) in relation to stress. Conclusions: Life in the period of the third wave of the pandemic was assessed by medical students as moderately satisfactory. Significant intensity of stress negatively correlated with life satisfaction. Health-promoting behaviors, and especially positive mental attitudes, seem to play a protective role in stressful situations and improve life satisfaction.

## 1. Introduction

The state of human health is influenced by many different elements, including genetic conditions, factors of the environment, and lifestyle. Lifestyle is shaped from early childhood and is the interaction of living conditions and personalized behavior patterns [1,2]. It can also be essential for both health and its quality. One of the elements closely related to health, which may have a significant impact on quality of life, is health behavior. These are conscious and intentional activities that take shape mainly in the second decade of life, but they are shaped throughout life [3]. Prohealth education commences at the stage of school education and is designed to develop correct habits that will bring a number of positive effects in later years. Health behaviors are created by internalizing information provided by schools, health workers, and the mass media, but the most important environment shaping this element of lifestyle is the family [4]. Negative patterns in the family, limited abilities to cope with increasing responsibilities and problems, and insufficient knowledge in the field of health may cause improper health behavior [5]. Undoubtedly, intensive teaching of proper attitudes in adolescence may decide the lifestyle in adulthood. One of the important periods in human life that may affect the formation of the above-mentioned attitudes is the time of study at a university [6]. First of all, the environment in which a young person begins to function changes diametrically. For most people, this is the beginning of independent life. Many people then change their place of residence; family relations and old acquaintances are lost, and completely new relationships are created, which may result in a specific environment, perpetuating wrong habits and an irregular lifestyle [5,7]. While studying at universities, there may be many dangers related to the potential risk of addiction, inadequate diet, lack of exercise, and inadequate coping with stress. It is an exceptionally specific period in which it is difficult to maintain proper health attitudes [8]. This is mainly due to an irregular lifestyle, a greater inclination to succumb to temptation, discovering the unknown, and the willingness to experience new aspects that create personality, identity, and worldview [5]. One of the next negative health factors that is clearly visible at this stage of life is stress. The period of studies is a time of credits, tests, and events that generate tension. The stress associated with this specific period of life has recently been exacerbated by the sudden change in the conditions of functioning that occurred with the announcement of the COVID-19 pandemic in March 2020 [9]. This moment became the beginning of a period of enormous changes that took place in the surrounding reality and which had not yet taken place in the twenty-first century [10,11]. In a very short time, a safe and relatively predictable life has been completely reorganized for almost everyone. The change in everyday work and learning, as well as the feeling of being in danger and uncertainty of the future, could significantly change the lifestyle and generate anxiety and fear [12]. The group particularly exposed to stress in this period were students of medical faculties [13,14,15]. In fact, medical studies in Poland last six years (twelve semesters), ending with the Medical Doctor (MD) title. After completing a 13-month-long postgraduate internship, students must pass the State Medical Examination to obtain a license to practice. What is important, clinical classes begin in the third year of study. It should be also noted that studying medicine is quite specific, and the students of medicine themselves are just as specific. The problem of high levels of stress in medical students has many causes. They are very ambitious young people, and the studies themselves cause a lot of stress during these years. It is also impossible not to see the causes of stress in the medical education system in Poland. Many students fear whether they have completed their studies, and often experience criticism from lecturers. Lack of support from lecturers, sometimes high inadequate knowledge requirements, and students’ feeling that the grades they receive are inadequate and unfair to their knowledge and time devoted to learning, especially when it intensifies in the first years of medical studies, contributes not only to the lowering of self-esteem, but also a high level of anxiety and stress [16]. During medical studies in Poland, classes were held exclusively in the hybrid mode till the onset of the pandemic. From this moment, the normal course of medical study changed diametrically. Difficulties related to the reorganization of academic science, as well as direct contact with patients during classes in clinical subjects, where the potential risk of COVID-19 infection was much higher than in other conditions, the need to exercise particular caution, wear protective clothing, and the constantly changing guidelines for procedures were factors generating tension and affecting health and life satisfaction [17,18]. Effective coping with stress in this period could be of particular importance for minimizing its negative effects, also in the long term [19]. Health-promoting behaviors are one of the elements that may influence the state of health and may also be an important protective element in situations of chronic stress [20,21].

The aim of the study was to evaluate prohealth behaviors as protective measures against stress during the third wave of the SARS-CoV-2 pandemic in a group of students of medicine at a Polish medical university. No other pieces of research investigating the relationship between health behaviors and the intensity of stress and life satisfaction during the COVID-19 pandemic have been found in the available scientific databases. It seems that health behaviors may be an important element influencing the health of individuals, especially in a situation as specific as the pandemic period, both through prophylactic behavior, which will be related to compliance with restrictions during this period, as well as through care for proper nutrition or physical activity. The importance of mental hygiene and proper mental attitudes should also be emphasized, because of their possible protective character.

The authors, based on the conviction of the importance of prohealth behaviors and the role of medical students in creating attitudes in the society, wanted not only to demonstrate their protective properties, but also the need to create positive attitudes towards them.

## 2. Materials

The study was conducted during the third wave of the pandemic (March–June 2021) among students of the Faculty of Medical Sciences in Zabrze, Medical University of Silesia. There were 306 participants, including 203 students in the 1st year of medicine at the age of 20.42 ± 1.92 years old and 103 students in the 5th year of medicine, aged 24.88 ± 1.7 years. The study was designed on the basis of the assumption of differences in the level of stress and health behaviors among students who have not started clinical classes yet (first year of study) and students who have had them for several years (5th year of study).

All respondents agreed to participate in the project. The response rate from the 1st-year students was 86%. The response rate from the 5th-year students was 78%. The sociodemographic characteristics of the respondents are presented in Table 1. Descriptive statistics of variables for the test group are presented in Table 2.

## 3. Methods

The following tools and psychometric questionnaires were used to assess the parameters tested:

### 3.1. Proprietary Demographic Data Questionnaire

This questionnaire was designed to control sociodemographic variables. It consisted of questions about gender, age, place of residence, close relationships, or level of education.

### 3.2. The Satisfaction with Life Scale (SWLS)

The scale by Diener, Emmons, Larson, and Griffin in the Polish adaptation of Juczyński (2001) allows for the study of the sense of satisfaction with life, understood as a subjective assessment of the quality of functioning. It contains five items. The respondent is asked to respond to each of the statements by specifying to what extent each of them relates to their life so far, from “strongly agree” (7 points) to “strongly disagree” (1 point). The grades are summed up, and the result obtained determines the degree of satisfaction with life. Scores are in the range of 5 to 35 points. They can be converted to standardized units on the sten scale. The results within 1–4 sten were considered low and within 7–10 sten as high. The results in the range of 5–6 sten correspond to average values. The reliability of the scale in Polish adaptation studies turned out to be adequate, the Cronbach’s alpha coefficient was 0.81 [20,22].

### 3.3. The Perceived Stress Scale (PSS-10)

The Perceived Stress Scale by Cohen, Kamarckmm, and Mermelstein, in the Polish adaptation of Juczyński and Ogińska-Bulik (2009), is a tool used to study the intensity of stress related to one’s own life situation. This assessment covers the past month. Stress is understood here as a reaction to difficult experiences. This tool consists of 10 items, to which the respondent refers to a five-point scale, where 0 means “never” and 4 “very often”. The subject takes an average of 5 min to complete the test. The scale score is between 0 and 40. The higher the score, the greater the stress level. The result is obtained by summing up the digits marked by the tested person in individual positions, while part of the results should be reversed according to the principle of 0 = 4, 1 = 3. The test results are normalized so that they can be presented using a sten scale. The tool has satisfactory psychometric properties in terms of accuracy and reliability. The Cronbach’s alpha coefficient for the entire tool was 0.86 [23].

### 3.4. Inventory of Health Behavior (IZZ)

This test, by Juczyński (2001), is a tool for measuring health behavior. It allows determining the general intensity of prohealth behaviors and four of its subscales—Proper Eating Habits (PEH), Preventive Actions (PA), Prohealth Activities (PhA), and Positive Mental Attitude (PMA). This tool consists of 24 statements, to which the respondent refers to a five-point scale, where 1 means “almost never”m and 5 means “almost always”. The possible overall score is in the range of 24–120 points. The higher the result, the greater the intensity of the declared prohealth behaviors. This ratio, after conversion into standardized units, is subject to interpretation on the sten scale. This test is the only tool in Polish cultural conditions that allows for a global assessment of health behaviors, involving the most important spheres of prohealth and preventive behaviors. The reliability of the tool for the entire test was 0.81, while for the individual subscales, it ranges between 0.60 and 0.80.

Due to the pandemic restrictions, the study was prepared in a Google form and was conducted in electronic form, using the Internet.

The prepared form contained information for participants about the aim of the study and discussed the filling instructions of tests used.

## 4. Statistical Analysis

Standard statistical procedures were used in the analyzes performed. The Mann–Whitney U test was used to assess the significance of differences between the studied groups. In order to assess the relationships between the data, the Spearman rank correlation coefficient was used. The evaluation of the influence of variables was analyzed by means of backward multiple stepwise regression. The significance level of *p* < 0.05 was adopted as statistically significant. The calculations were made in Statistica version 13.3.

### Ethical Consideration

The Bioethics Committee of the Medical University of Silesia approved the research (PCN/0022/KBI/66/2).

Students were informed about the anonymity and confidentiality of the research. What is more, they were informed that they could stop the study whenever they wanted. Information about the study and informed consent were included in the first part of the prepared form.

Students or researchers were not offered any compensation as an incentive to participate. The authors received no specific funding for this study.

## 5. Results

### 5.1. Life Satisfaction

When analyzing the results obtained in the study group in the SWLS, an average result of 23.22 ± 5.97 points was obtained, which, when converted into stens, allows estimating the level of life satisfaction as medium–high. After dividing into groups, the result was 23.48 ± 6.05 points, respectively. For 1st-year and 24.01 ± 5.85 points for 5th-year students, the difference was not statistically significant (Table 3).

### 5.2. The Perceived Stress

When analyzing the results obtained in the study group in PSS-10, an average score of 21.10 ± 5.89 points was obtained, which, when converted into sten values, allows the stress level to be determined as high. When analyzing the results for both groups separately, 22.22 ± 5.71 points were obtained for 1-year and 18.88 ± 5.62 for 5th-year students, respectively. The difference was statistically significant (*p* < 0.000) (Table 3).

### 5.3. Health Behaviors

When analyzing the results obtained in the group studied in the IZZ, an average result of 80.06 ± 13.81 points was obtained, and after division into groups, among 1st-year students, 78.65 ± 13.38 and, in the group of 5th-year students, 82.85 ± 14.27. Converted to sten values, all three results are at an average level. The groups differed statistically (*p* = 0.016) in terms of the overall IZZ results and in the subscale of Proper Eating Habits (PEH) (*p* = 0.016). In both cases, a better result was obtained in the group of fifth-year students (Table 3).

### 5.4. Analysis of Relationships between the Tested Parameters

The analysis of the relationship between the tested parameters was presented for both groups tested together. There were statistically significant negative correlations between the results obtained in the Health Behavior Inventory of IZZ and the results obtained on the PSS-10 stress scale and positive correlations with the results obtained on the SWLS life satisfaction scale. Taking into account the components of health behaviors, statistically significant positive correlations with the results obtained on the life satisfaction scale were obtained in all subscales of the IZZ inventory. Moreover, the results obtained on the PSS-10 stress scale were significantly negatively correlated with the results obtained in the subscale of Positive Mental Attitudes (PMA) and Prohealth Activities (PhA) and positively with the results in the subscale of Preventive Actions (PA) (Table 4). In order to assess the impact of prohealth behavior parameters on the perceived stress and life satisfaction, the multiple regression method was used, and the positive effect of a Positive Mental Attitude (PMA) on life satisfaction was demonstrated, as well as the protective effect of PMA and Prohealth Activities (PhA) in relation to perceived stress (Table 5).

## 6. Discussion

Research on the determinants of health, which has been carried out for many years, allowed concluding that the greatest importance, determined at the level of 50–60%, for maintaining its proper condition is lifestyle and health behaviors [23]. These include the spheres of physical, mental, and social health and preventive behaviors. Behaviors related to physical health include taking care of the body, level of physical activity, rational nutrition, proper working conditions, and sleep. Behaviors related mainly to mental and social health are related to, inter alia, maintaining social contacts and coping with permanent stress. In turn, preventive actions include: self-monitoring of health, undergoing preventive examinations, and safe behavior in everyday life [24]. Health behaviors may play a greater or lesser role in minimizing the effects of stress. The COVID-19 pandemic has become a special period that could generate stress in almost every human being [25,26,27]. In our study in a group of medical students, the level of stress on the PSS-10 scale was 21.10 ± 5.89 points and was slightly lower than that presented in Rogowska’s research conducted in a similar period and in the study group of students with 23.53 ± 4.76 points [28]. Rutkowska’s research conducted at the same time showed a stress level of 20.85 ± 5.63 [29]. We conducted our study to assess the perceived stress in two groups of students of one medical faculty but at different stages of life, taking into account that the first group was just starting their education and the second group was almost at its end. This factor could probably be one of the determinants of the differences in perceived stress, which was more intense among students in the first year of studies (22.22 ± 5.71 points) and was comparable to the level reported in the group of nurses working with patients infected with SARS-CoV-2 (22.22 ± 5.94 points) [30]. Stress was more intense among students in the first year of studies. This is a very interesting result, but this also raises many questions. Was this stress caused only by pandemics? In Poland, medical students most often resign or are removed after the first year. They are suddenly in a very competitive environment, where only the best students are. Authors consider that other factors, except the pandemic situation, could influence this higher level of stress in the first year of medical studies, such as lack of clinical experience, uncertainty about the future, and fear of being removed from studies.

The opposite results mentioned above were obtained in a study conducted in March 2021 by Ali, where the intensity of stress among 6th-year students was 34.41 ± 7.77 points, and it was much higher than among 2nd-year students (30.00 ± 6.59 points) [31]. In the Aslan study, however, no differences were found between the severity of stress depending on the year of study [32]. It seems that people who started their education in completely new conditions of reality, significantly changed by the COVID-19 pandemic, are a group more vulnerable to experiencing stress due to their younger age, less educated coping mechanisms, less knowledge about the world, and less social support resulting mainly from a change in the environment that occurs at this stage of life. In this period, classes were held in a hybrid manner; some classes were online and some stationary, which could also be one of the causes of stress. Students did not have the time and opportunity to establish new relationships and integrate, which usually takes place in the first months of the first year of studies [33].

The fifth-year students were in a much better situation, having spent several years in a group together and a certain amount of knowledge about how to function at the university. Although the fifth-year students have many more possibilities of contact with patients and the first-year students have almost no clinical classes, this did not affect the perceived stress. Despite differences in the intensity of perceived stress, the study groups did not differ in terms of their satisfaction with life. Satisfaction with life in the group of 1st-year students was 23.46 ± 6.05 points and, for the 5th-year students, 24.01 ± 5.84 points. In both cases, it was higher than in the research conducted by Rogowska, where the result obtained in a group of 605 students was 19.78 ± 7.00 points, Refs. [28,34], and in the Lopes study from March 2020, where the satisfaction was 20.00 ± 0.85 points. [35], as well as in the Turkish research by Aslan (16.72 ± 6.81 points) [32]. Despite the difference in the level of stress experienced by students in the first and fifth year of studies and a similar level of life satisfaction, a negative relationship was found between the level of perceived stress and the level of life satisfaction in both study groups. The above-mentioned research studies also indicated such a relationship [28,32,35]. The period of the pandemic, especially in its initial stage, was also a time of intense changes in lifestyle. Periodic lockdown, staying in quarantines, significant limitation of the possibility of visiting gyms or fitness clubs, and in some moments even leaving the house, except in cases of special necessity, led to the possibilities of physical activity being severely limited [36,37]. Access to health care was also limited, and almost all its forces were devoted to fighting the pandemic [38]. External activity limitations were also often associated with a change in eating habits, which could take place in two directions. More time could transform into greater care for the regularity and quality of meals; on the other hand, frustration caused by the inability to function normally, stress related to the sudden closure of all household members, often on a rather small surface, and the proximity of food sources could be conducive to an improper diet [39,40]. Health behaviors acquired earlier in life in this specific situation could have become a protective factor against stress and had an impact on the assessment of life satisfaction. In our research, in the overall assessment of prohealth behaviors, an average score of 80.06 ± 13.81 points was obtained, which, when converted into stens, allows us to estimate health-related behaviors at an average level. This indicator was slightly lower than the normative one (81.82 points) obtained in a study on a population of 496 adults [20]. On the other hand, when analyzing the behavior in individual groups of respondents, a statistically significantly better result was obtained in the group of students in the 5th year (82.85 ± 14.27 points) compared to students in the 1st year (78.65 ± 13.38 points). This is probably due to greater awareness of the importance of health-related behaviors associated with greater knowledge and experience among older respondents. When analyzing the values obtained in individual subscales, it can be observed that the respondents obtained the highest score on the scale of Preventive Actions, which, in this period of increased emphasis on care for all methods of preventing infections, does not seem to be a surprising fact. The statistical differences between the groups concerned only Proper Eating Habits, which turned out to be better among fifth-year students. In the available scientific reports, groups of students of medical faculties do not rank well in the assessment of prohealth behaviors. In the research conducted by Radosz comparing students of obstetrics, physiotherapy, and nursing, the group of nursing students fared best, reaching 78.12 ± 11.92 points. In Ref. [41] and in the Kulik research conducted among psychology students, health behaviors in the 1st year of studies amounted to 77.22 ± 13.99 points and 76.61 ± 14.15 points in the same group after 4 years of study [42].

Taking into consideration the relationship between health behaviors and perceived stress, a negative correlation with the perceived stress and a positive correlation with life satisfaction were shown both in the whole group and in the analysis broken down into groups. Similar results were presented in Piech’s research, where levels of health behaviors were negatively correlated with a high level of anxiety in the group of women in the postpartum period [43], and the studies by Nowak et al. [44] emphasized that depressed people showed less care for their health. In Luis’ studies, similar correlations were obtained, despite the use of other scales: for the assessment of life satisfaction, Ryff’s Psychological Well-Being Scale-29, and for the assessment of health behaviors, Self-Care Activities Screening Scale-1 [45]. By analyzing the relationship between the components of health behaviors and the severity of stress, negative relationships between perceived stress and health practices and mental attitudes were shown and positive relationships with preventive behaviors for both groups of respondents as a whole. Preventive behaviors during this particular period could be associated with increased stress for various reasons. During the pandemic, access to health care was significantly limited; therefore, it became difficult to conduct any preventive examinations during this period [46,47,48,49]. On the other hand, constant reminding and emphasizing the need to maintain social distance, wearing masks, and frequent hand washing could generate stress by the very fact of the enormous importance that these behaviors acquired during this time [50,51]. In the assessment of the relationship of individual components of health behaviors with life satisfaction, positive correlations of all components of health behaviors with life satisfaction were shown, but only with regard to a positive mental attitude was a significant positive effect on the parameter under study (b = 0.04). What is important, life satisfaction can also depend on the individuals’ personality traits. We obtained similar results in our study, assessing health-promoting behavior during the COVID-19 pandemic in a group of medical personnel [52]. Investments in shaping proper health behaviors may have a significant impact on the health of society in the coming years. Proper health behaviors developed in early adolescence will be an important element in determining health in its later stages, and they can also be a protective factor against stress and its long-term effects. The chronic nature of the pandemic period may be a potential source of long-term stress consequences.

It would seem crucial to develop an attitude among students, making them aware of the importance of their profession as a kind of guide and example for society. According to the authors, an example for young people should be given by their mentors. Medical students could, and even should become, social managers, not only because of the protective importance of health behaviors, but also to create attitudes in their environment. Students could spread reflection both on their own and in any other company. The key is to make sense of such activities among this target group. Developing health attitudes among medical students is not only a benefit for themselves, but also for the environment, following the normative script, which defines three levels of rewards: personal; family level; and, the broadest, community level [53].

Medical students are a group that will have a great influence on creating proper health attitudes in the future, especially in the group of their future patients. Knowing about the types of health-promoting behaviors and ways of modifying them can largely contribute to improving the health of the society, as well as better functioning in periods of increased stress, such as the pandemic period [54].

## 7. Study Limitations

The presented study, like any other study in which results were collected using the Internet, has certain limitations. First of all, collecting data via the Internet form made it impossible to control the course of filling in the questionnaires. The limitation of the article is also collecting the research material in one unit, without creating comparative groups from other teaching units, which requires supplementing as part of the continuation of the research. The lack of similar research in medical databases turned out to be a major limitation and thus a problem with relating to the results. Additionally, results based on the study group from one university seem to be a problem, especially in terms of result generalization. Due to the length of the article, the study also did not describe the significance of differences in the scope of the studied variables resulting in the study group from sociodemographic differences, although the authors are aware of that. What is more, a helpful study could be the evaluation of the level of stress among first-year medical students from several universities in Poland depending on different psychological and sociodemographic factors.

## 8. Conclusions

Medical workers, and also medical students, play an important role in publicly demonstrating commitment to the full range of health-supporting behaviors. In the course of this study, a positive relationship between life satisfaction and health behaviors and a negative relationship with stress during the COVID-19 pandemic in medical students were confirmed. Health-promoting behaviors, especially Positive Mental Attitudes, may play a protective role in a situation of increased stress related to the COVID-19 pandemic. The development and implementation of adequate programs dedicated to prevention and intervention at universities should be one of the priorities in the fight against the COVID-19 pandemic. The results presented the need to promote health behaviors in the current situation in order to reduce the long-term negative consequences of the pandemic in the field of mental health. Moreover, there is a great need to educate prohealth attitudes among medical students so that they can become role models for their relatives and the environment.

## Figures and Tables

**Table 1 ijerph-19-06706-t001:** Demographic characteristics of the sample.

Variables	Total (*n* = 306)	1st Year (*n* = 203)	5th Year (*n* = 103)
f (rf)	f (rf)	f (rf)	
**Gender**			
Men	116 (37.9%)	73 (36.0%)	43 (42.0%)
Women	190 (62.1%)	130 (64.0%)	60 (58.0%)
**Marital Status**			
Single	183 (59.8%)	137 (67.5%)	46 (44.7%)
In relationship	123 (40.2%)	66 (32.5%)	57 (55.3%)
**Place of Residence**			
Urban	225 (73.5%)	139 (68.5%)	86 (83.5%)
Rural	81 (26.5%)	64 (31.5%)	17 (16.5%)
**Chronic Diseases**			
No	242 (79.1%)	171 (84.0%)	71 (68.9%)
Yes	64 (20.9%)	32 (16.0%)	32 (31.1%)

**Table 2 ijerph-19-06706-t002:** Descriptive statistics of variables for the test group.

*n* = 306	Mean	Standard Deviation	Median	Min.	Max.	PU −95%	PU +95%
**SWLS**	23.666	5.979	25.000	5.000	35.00	5.540	6.495
**PSS**	21.101	5.893	21.000	7.000	33.000	5.460	6.400
**IZZ**	80.06	13.812	80.000	43.000	120.000	12.798	15.000
**PEH**	19.205	5.083	19.000	6.000	30.000	4.710	5.521
**PhA**	21.003	4.414	21.000	9.000	30.000	4.089	4.794
**PMA**	20.081	4.589	20.000	7.000	30.000	4.251	4.984
**PA**	19.774	4.359	20.000	9.000	30.000	4.039	4.735

SWLS—The Satisfaction with Life Scale, PSS—Perceived Stress Scale, IZZ—Health Behavior in general, PEH—Proper Eating Habits, PMA—Positive Mental Attitude, PhA—Prohealth Activities, PA—Preventic Actions.

**Table 3 ijerph-19-06706-t003:** Differences between 1st- and 5th-year students.

	1st Year (*n* = 203)			5th Year (*n* = 103)	
	Mean	Standard Deviation	Median	Mean	Standard Deviation
**SWLS**	23.484	5.514	24.000	24.019	5.849
**PSS**	22.226	5.718	23.000	18.883	5.620
**IZZ**	78.650	13.384	79,000	82,854	14,279
**PEH**	18.684	4.916	18.000	20.233	5.273
**PhA**	20.679	4.344	21.000	21.640	4.502
**PMA**	19.842	4.140	20.000	20.553	4.662
**PA**	19.443	4.475	20.000	20.427	4.064

SWLS—The Satisfaction with Life Scale, PSS—Perceived Stress Scale, IZZ—Health Behavior in general, PEH—Proper Eating Habits, PMA—Positive Mental Attitude, PhA—Prohealth Activities, PA—Preventic Actions.

**Table 4 ijerph-19-06706-t004:** Relationships between life satisfaction and perceived stress and health behaviors.

*n* = 306	SWLS	PSS	IZZ	PEH	PhA	PMA	PA
**SWLS**	1.000	−0.490 ***	0.385 ***	>0.117 **	0.250 ***	0.534 ***	0.268 ***
**PSS**		1.000	−0.309 ***	−0.050	−0.073	−0.506 ***	−0.330 ***
**IZZ**		1.000	0.724 ***	0.736 ***	0.771 ***	0.695 ***	
**PEH**		1.000	0.452 ***	0.327 ***	0.312 ***		
**PhA**		1.000	0.452 ***	0.317 ***			
**PMA**		1.000	0.528 ***				
**PA**		1.000					

SWLS—The Satisfaction with Life Scale, PSS—Perceived Stress Scale, IZZ—Health Behavior in general, PEH—Proper Eating Habits, PMA—Positive Mental Attitude, PhA—Prohealth Activities, PA—Preventic Actions, BDI—depression, * *p* < 0.05, ** *p* < 0.01, *** *p* < 0.001.

**Table 5 ijerph-19-06706-t005:** Regression models of life satisfaction and perceived stress in light of health behaviors.

	Βeta β	Βeta β SE	Beta β	Beta β SE	*p*
**LIFE SATISFACTION**					
**Constant**	---	---	9.928	1.310	**0.000 ***
**PMA**	0.525	0.049	0.684	0.064	**0.000 ***
R^2^corr. = 0.273; F (1.304) = 115.700; *p* < 0.000; SEE = 5.098					
**PERCEIVED STRESS**					
**Constant**	---	---	33.768	1.701	**0.000 ***
**PMA**	−0.461	0.062	−0.592	0.080	**0.000 ***
**PhA**	−0.186	0.058	−0.252	0.079	**0.002 ***
**PA**	0.150	0.055	0.200	0.073	**0.006 ***
R^2^corr. = 0.278; F (3.302) = 40.151; *p* < 0.000; SEE = 5.007					

SE—standard error, R^2^corr.—corrected R-squared, SEE—error of estimation, PMA—Positive Mental Attitude, PhA—Prohealth Activities, PA—Preventic Actions, * statistically significant at *p* < 0.05.

## Data Availability

The study did not report any data.

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
