# Peer review of "Life Satisfaction and Perceived Stress versus Health Promoting Behavior among Medical Students during the COVID-19 Pandemic"

_ijerph, 2022, doi:10.3390/ijerph19116706_

Round 1
Reviewer 1 Report
- The article satisfactorily explains the theoretical frameworks - both of "life satisfaction", "health promotion behavior", etc.
- In my opinion, there is a need for a prologue (another summary - after the existing summary) that will come as a critical-personal reflection or expression of the writers - who will make their own theoretical contribution - in relation to the "ideal script" proposed by them to promote positive environment or health promotion behavior among medical students (both during an epidemic crisis and on an ongoing basis, during a routine). after all, this is the main motivation that readers will have for reading this article - its contribution to the care and building of this type of environment.
2. In my opinion, the authors should to use the term "Normative Script" which can be is very essential - and discuss how to create expectations in which it will be clear to any medical student that as part of the normative script/scenario/expectations of his development - he must act and behave as an agent of health promotion behavior among colleagues and patients.
For the term "Normative Script" see:
Lebel, U. and Masad, D. “Life Scripts, Counter Scripts and Online Videos: The Struggle of Religious-Nationalist Community Epistemic Authorities against Military Service for Women”. Religions 12(9), 2021, 750
Author Response
Dears Editor and Reviewers,
Thank you for revising article entitled “Life satisfaction and perceived stress versus health promoting behavior among medical students during the Covid-19 pandemic” We have reorganized the paper following Reviewers’ comments. All changes in the text of article have been written in blue pencil. Our responds to the Reviewers are presented below.
The article satisfactorily explains the theoretical frameworks - both of "life satisfaction", "health promotion behavior", etc. In my opinion, there is a need for a prologue (another summary - after the existing summary) that will come as a critical-personal reflection or expression of the writers - who will make their own theoretical contribution - in relation to the "ideal script" proposed by them to promote positive environment or health promotion behavior among medical students (both during an epidemic crisis and on an ongoing basis, during a routine). after all, this is the main motivation that readers will have for reading this article - its contribution to the care and building of this type of environment.
We tried to explain and describe problem mentioned above in Discussion section .
In my opinion, the authors should to use the term "Normative Script" which can be is very essential - and discuss how to create expectations in which it will be clear to any medical student that as part of the normative script/scenario/expectations of his development - he must act and behave as an agent of health promotion behavior among colleagues and patients.
For the term "Normative Script" see: Lebel, U. and Masad, D. “Life Scripts, Counter Scripts and Online Videos: The Struggle of Religious-Nationalist Community Epistemic Authorities against Military Service for Women”. Religions 12(9), 2021, 750
We tried to present, taking into account mentioned article, how to promote health behaviors among medical students and how to create proper attitudes that can be presented to their colleagues and future patients.
We would like to thank You once again for your valuable review.
Authors

Reviewer 2 Report
Thank you for the possibility to read this interesting article. My suggestions for the authors are below:
Abstract
Please delate the sentence: “Among the statistical methods, the Mann- 19 Whitney U test, the Spearman's rank correlation coefficient, and the backward multiple step regression procedure were used.” It is unnecessary. You should clearly state here that the survey was done online.
Introduction
I think that the authors omitted two important threads in the introduction. 1) How does the education of medical students in Poland looks like? How many years? When does clinical start, etc? How the classes during your study was conducted only online, hybrid? See, how it is described here: A Study of Differences in Compulsory Courses Offering Medicine Humanization and Medical Communication in Polish Medical Schools: Content Analysis of Secondary Data - PMC (nih.gov); 2) that the study of medicine is very specific, medical students are specific - they are very ambitious young people, and the studies themselves cause a lot of stress. See, A Qualitative Study of the Mistreatment of Medical Students by Their Lecturers in Polish Medical Schools - PMC (nih.gov).
Please add research question/s after the aim.
Materials and Methods
Are the data from the study publicly available?
How many students are in the Faculty of Medical Sciences in Zabrze? Why only two years? Why those years?
What was the response rate?
Discussion
Stress was more intense among students in the first year of studies. This is a very interesting result, but this also rise the questions. How this is possible? Do they have any contact with the patients? Was this stress caused only by pandemics? In Poland, medical students most often drop out after the first year… They are suddenly in a very competitive environment, where only the best students are. I am sure you should also consider the other factors that could influence this higher level of stress in the 1st year.
What further research can be done to help clarify or expand on the threads of this study?
Conclusion
I think the conclusions should be developed. Such a summary of points is insufficient.
Author Response
Abstract
Please delate the sentence: “Among the statistical methods, the Mann- 19 Whitney U test, the Spearman's rank correlation coefficient, and the backward multiple step regression procedure were used.” It is unnecessary.
The sentence was deleted.
You should clearly state here that the survey was done online.
The statement has been provided in the text of article.
Introduction
I think that the authors omitted two important threads in the introduction.
1) How does the education of medical students in Poland looks like? How many years? When does clinical start, etc?
How the classes during your study was conducted only online, hybrid?
See, how it is described here: A Study of Differences in Compulsory Courses Offering Medicine Humanization and Medical Communication in Polish Medical Schools: Content Analysis of Secondary Data - PMC (nih.gov);
Information have been implemented in the text of article.
2) that the study of medicine is very specific, medical students are specific - they are very ambitious young people, and the studies themselves cause a lot of stress. See, A Qualitative Study of the Mistreatment of Medical Students by Their Lecturers in Polish Medical Schools - PMC (nih.gov).
Please add research question/s after the aim.
We have done it.
Materials and Methods
Are the data from the study publicly available?
Data from the study are available at the first author of the article.
How many students are in the Faculty of Medical Sciences in Zabrze? Why only two years? Why those years?
What was the response rate?
Information have been completed in the text of manuscript.
Discussion
Stress was more intense among students in the first year of studies. This is a very interesting result, but this also rise the questions. How this is possible? Do they have any contact with the patients? Was this stress caused only by pandemics? In Poland, medical students most often drop out after the first year… They are suddenly in a very competitive environment, where only the best students are. I am sure you should also consider the other factors that could influence this higher level of stress in the 1st year.
Information have been completed in the text of Discussion section.
Conclusion
I think the conclusions should be developed. Such a summary of points is insufficient.
We tried to develop conclusions.

Reviewer 3 Report
Dear authors,
The authors conducted a study that aimed to evaluate health promoting behaviors as a factor supporting life satisfaction and a protective factor against stress in a group of Polish medical students during the third wave of the SARS-CoV-2 pandemic. For that the authors used a sample of medical students.
Despite the positive features of the study, there are some considerations to take care before the paper is published.
The following comments will summarize my appreciation and major concerns with your paper. I hope these comments help you further improve your study.
Theoretical background
- Page 2, line 57 change loosened for lost.
- The literature review is adequate and suits the main goals of the paper.
- What are the main goals of the paper? What do you want to achieve? And what do you add to the literature.
Method
- Page 2 line 91 change people for participants.
- Did the authors control for any variables?
- Page 3 line 106 is missing the year of reference for the adapted SWLS.
- The same for page 3, line 117 for the adapted version of the PSS.
- The same for page 4, line 129.
- Why did the authors use different Liker scales? #Why not collecting data with equal measurement points?
- Please add more information regarding the data collection procedure. How did the students were recruited? What is the response rate?
Discussion
The discussion was short and did little more than summarise your findings.
- Please, develop the discussion section, considering the theoretical implications.
- The limitations and future research should be elaborated.
- What are the main practical implications of the study? And what do you add that is not tested before?
Author Response
Theoretical background
1.Page 2, line 57 change loosened for lost. – it was corrected
5. What are the main goals of the paper? What do you want to achieve? And what do you add to the literature.
Information on above suggestions have been provided in Introduction.
Method
- Page 2 line 91 change people for participants. – it was corrected
- Did the authors control for any variables? –
Group differences, taking into consideration sociodemographic variables, were examined. The groups did not differ in terms of sex and place of residence. Due to the length of work, the above data was not presented in the article.
- Page 3 line 106 is missing the year of reference for the adapted SWLS. – it was completed
- The same for page 3, line 117 for the adapted version of the PSS. - it was completed
- The same for page 4, line 129. it was completed
- Why did the authors use different Liker scales? #Why not collecting data with equal measurement points?
We used the tools according to their original layout. Changing the scale of responses could adversely affect the way of giving answers. We conducted the examination in agreement with original tests instructions.
Please add more information regarding the data collection procedure. How did the students were recruited? What is the response rate? –
All students received a link to the study via the university's information systems.
Discussion
The discussion was short and did little more than summaries your findings.
- Please, develop the discussion section, considering the theoretical implications.
The Discussion section was developed.
- The limitations and future research should be elaborated.
The limitations of the study were elaborated.
- What are the main practical implications of the study? And what do you add that is not tested before?
Discussion section was completed with practical implications mentioned above.

Reviewer 4 Report
The paper concerns an actual topic of coping with Covid-19 pandemic among medical workers, focusing on the differences between fresh and graduate medical students. We do not have yet any large-scale picture of psychological factors maintaining the resilience when one faces the challenges of such stressful life conditions. The neglecting of Sars-Cov-2 protective measures undoubtedly has various psychological reasons at its core. But medical workers – and students, of course - play an important role in publicly demonstrating commitment to the full range of health-supporting behaviours. The research in students, generally, allows to extrapolate the facts obtained to a wider audience of young people whose social circumstances change radically after transition from secondary school to adult life in university campus. This transition enforces the stress influenced by the pandemic counter-efforts. That is why the research and combination of psychometric tools are interesting and quite new in relates with the old idea of stress influencing health promoting behaviour. The correlations in combination with multiple regression method demonstrate pragmatically useful structure of health-promoting behaviour delineating its critical elements in relation with Covid-19 stress. In Discussion section the authors give satisfactory explanations of the discrepancies between their empirical data and the facts obtained in other research on the problem: they found a generation that started their adult life in the campus under the restrictive counter-Sars-Cov-2 measures. I find the article may be published in the journal as it is.
Author Response
Dears Editor and Reviewers,
Thank you for revising article entitled “Life satisfaction and perceived stress versus health promoting behavior among medical students during the Covid-19 pandemic” We have reorganized the paper following Reviewers’ comments. All changes in the text of article have been written in blue pencil.
We would like to thank you for your positive opinion. We do appreciate your effort and work devoted to our manuscript.

Reviewer 5 Report
This research attempted the relationship between life satisfaction and stress vs health promoting. It is quite interesting in this COVID-19 pandemic. The results are interesting, too!!
However, "life satisfaction" is also influenced by the personality. So to speak, life satisfaction depends on the individual personality. Please add your consideration on this point in the discussion!
Author Response
Dears Editor and Reviewers,
Thank you for revising article entitled “Life satisfaction and perceived stress versus health promoting behavior among medical students during the Covid-19 pandemic” We have reorganized the paper following Reviewers’ comments. All changes in the text of article have been written in blue pencil.
We would like to thank you for your positive opinion. We do appreciate your effort and work devoted to our manuscript. Discussion has been completed with information on personality traits and their connection with satisfaction of life.

Round 2
Reviewer 2 Report
I have only minor comments on the manuscript. It seems to me that the authors did not take into account all my comments about the introduction (describing Polish medical students as ambitious, stressed /adding research question/s) and about the conclusion (points).